# High-Throughput Microfluidic Real-Time PCR for the Detection of Multiple Microorganisms in Ixodid Cattle Ticks in Northeast Algeria

**DOI:** 10.3390/pathogens10030362

**Published:** 2021-03-18

**Authors:** Ghania Boularias, Naouelle Azzag, Clemence Galon, Ladislav Šimo, Henri-Jean Boulouis, Sara Moutailler

**Affiliations:** 1Research Laboratory for Management of Local Animal Resources, Higher National Veterinary School of Algiers, Rue Issad Abbes, El Alia, Algiers 16025, Algeria; ghania.boularias@gmail.com; 2Laboratoire de Santé Animale, Ecole Nationale Vétérinaire d’Alfort, UMR BIPAR, ANSES, INRAE, F-94700 Maisons-Alfort, France; clemence.galon@anses.fr (C.G.); ladislav.simo@vet-alfort.fr (L.Š.); henri-jean.boulouis@vet-alfort.fr (H.-J.B.)

**Keywords:** Algeria, ixodid ticks, tick-borne pathogens, co-infection, cattle, high-throughput microfluidic real time PCR

## Abstract

Ixodid ticks are hematophagous arthropods considered to be prominent ectoparasite vectors that have a negative impact on cattle, either through direct injury or via the transmission of several pathogens. In this study, we investigated the molecular infection rates of numerous tick-borne pathogens in ticks sampled on cattle from the Kabylia region, northeastern Algeria, using a high-throughput microfluidic real-time PCR system. A total of 235 ticks belonging to seven species of the genera *Rhipicephalus*, *Hyalomma*, and *Ixodes* were sampled on cattle and then screened for the presence of 36 different species of bacteria and protozoans. The most prevalent tick-borne microorganisms were *Rickettsia* spp. at 79.1%, followed by *Francisella*-like endosymbionts (62.9%), *Theileria* spp. (17.8%), *Anaplasma* spp. (14.4%), *Bartonella* spp. (6.8%), *Borrelia* spp. (6.8%), and *Babesia* spp. (2.5%). Among the 80.4% of ticks bearing microorganisms, 20%, 36.6%, 21.7%, and 2.1% were positive for one, two, three, and four different microorganisms, respectively. *Rickettsia aeschlimannii* was detected in *Hyalomma marginatum*, *Hyalomma detritum*, and *Rhipicephalus bursa* ticks. *Rickettsia massiliae* was found in *Rhipicephalus sanguineus*, and *Rickettsia*
*monacensis* and *Rickettsia helvetica* were detected in *Ixodes*
*ricinus*. *Anaplasma marginale* was found in all identified tick genera, but *Anaplasma centrale* was detected exclusively in *Rhipicephalus* spp. ticks. The DNA of *Borrelia* spp. and *Bartonella* spp. was identified in several tick species. *Theileria orientalis* was found in *R. bursa*, *R. sanguineus*, *H. detritum*, *H. marginatum*, and *I. ricinus* and *Babesia bigemina* was found in *Rhipicephalus annulatus* and *R. sanguineus*. Our study highlights the importance of tick-borne pathogens in cattle in Algeria.

## 1. Introduction

Ixodid ticks are blood-sucking arthropods, and are considered to be prominent vectors of pathogens for humans as well as domestic and wild animals. They are known to transmit a wide variety of causative agents such as bacteria, protozoa, and viruses that may subsequently infect the mammal host. Globally, the incidence of tick-borne diseases is growing, mostly due to increased interactions between pathogens, vectors, and hosts [1,2]. Furthermore, climate change, including the prolongation of seasons, global warming, and changing precipitation patterns, extends the geographic range of a number of tick species and the pathogens they carry [3].

Ticks are regarded as the primary vectors of pathogens affecting livestock [4] and up to 80% of cattle worldwide are at risk of coming into contact with ticks and contracting diseases caused by transmitted tick-borne pathogens (TBPs) [5]. An individual animal can be infested with hundreds or even thousands of ticks, which clearly magnifies their effect on the host, either by direct injury or by the transmission of pathogens [4]. In addition, the co-transmission of several pathogens may lead to co-infection in animals, which aggravates their vital prognosis or, in some cases, gives rise to atypical forms, thereby complicating diagnosis [6]. Thus, it is very important to detect and identify TBPs, so that veterinarians can predict the risk of infection and subsequently implement appropriate control measures.

Hard ticks belonging to the genera *Hyalomma*, *Rhipicephalus*, *Ixodes*, and *Haemaphysalis* have been identified feeding on grazing cattle in Algeria [7]. In addition, various studies using classical molecular techniques have detected multiple pathogens in these ticks, e.g., the genera *Rickettsia*, *Anaplasma*, *Coxiella*, and *Theileria*, in cattle ticks from Algeria [8,9,10,11]. All of these studies used classical molecular methods that can only detect a few pathogens simultaneously, are time-consuming, and require large volumes of DNA for the detection of multiple pathogens. Moreover, evidence of non-pathogenic commensal microorganisms called endosymbionts is poorly documented in Algeria but provides useful information because they may influence the transmission of other tick-borne microorganisms or become pathogenic for humans and/or animals [6,12,13].

To do so, microfluidic-based high-throughput PCR systems have been described by various studies as the most sensitive approach to detect TBPs [6,14,15]. These systems allow the rapid and simultaneous detection of numerous microorganisms using a small volume of DNA, thereby making it possible to carry out large-scale epidemiological investigations on TBPs in ticks [14]. Here, we successfully employed this approach to investigate the distribution of TBPs in bovine ticks from the Kabylia region of Algeria for the first time.

## 2. Results

### 2.1. Taxonomical Identification of Collected Tick Species

A total of 518 male and 537 female hard ticks were manually detached from bovines. The females varied in size due to different feeding durations. We did not find any immature tick stages in the collected samples. Three common tick genera were identified: *Rhipicephalus* (646/1055, 61.2%), *Hyalomma* (390/1055, 36.9%), and *Ixodes* (19/1055, 1.8%). Among these genera, seven different hard tick species were recognized. *R. bursa* and *H. detritum* were the two most common species, followed by *H. marginatum* and *H. lusitanicum* and finally *I. ricinus*, *R. sanguineus*, and *R. annulatus*. For morphologically deformed *Hyalomma* and *Rhipicephalus* ticks, specimens were determined to the genus level only. The difference in frequencies among identified species is described in Figure 1. In addition, co-infestation with different species of ticks was observed on the sampled bovines.

### 2.2. Infection Rates of Microorganisms and Their Co-Infection Rates in Ticks

Among all investigated ticks (Table 1), six pathogen genera were identified as follows: *Rickettsia* (79.1%, 186/235), *Theileria* (17.8%, 42/235), *Anaplasma* (14.4%, 34/235), *Bartonella* (6.8%, 16/235), *Borrelia* (6.8%, 16/235), and *Babesia* (2.5%, 6/235). The overall rate of *Francisella*-like endosymbionts (FLE) was 62.9% (148/235), with a positivity rate of 88.5% (100/113) for *Hyalomma* spp., followed by 81.3% (11/13) for *Ixodes* and finally 33.9% (37/109) for the genus *Rhipicephalus*. Neither *Coxiella* spp. nor *Hepatozoon* spp. were detected in any ticks.

Among the *Rickettsia*-tested ticks, 11.5% (27/235) were positive for *R. aeschlimannii* and 4.8% (11/235) for *R. massiliae*, and a total of 26.3% (62/235) of ticks were positive for undetermined *Rickettsia* spp.; 10 *Rickettsia* specimens were chosen randomly and sequenced. The BLAST search on these 10 specimens (confirmed by *gltA* gene amplification) revealed that five showed 100% identity with *R. monacensis* (Accession nos. JX040640.1 and KJ663735.1), two showed 100% identity with *R. helvetica* (Accession no. KY231199.1), two showed 100% identity with uncultured *Rickettsia* sp. (Accession no. KU596570.1) and one showed 81.8% identity with another uncultured *Rickettsia* sp. (Accession no. AP019865.1). For the 36.5% (86/235) of positive samples harboring multiple *Rickettsia* species at the same time, we sequenced 10 specimens and the BLAST results indicated 100% identity with an uncultured *Rickettsia* sp. (Accession no. KU596570.1).

*R. aeschlimannii* was detected in *H. marginatum*, *H. detritum* and *R. bursa* ticks. Four out of six *R*. *sanguineus* ticks were positive for *R. massiliae. R. massiliae* was also amplified in *Hyalomma* ticks. *R. monacensis* and *R. helvetica* were detected only in *I. ricinus* ticks (Table 1).

DNA of *Theileria* spp. was detected in 17.8% (42/235) of samples, and nested PCRs followed by sequencing showed an identity of 100% with *T. orientalis* (Accession no. MH208641.1). This species was detected in *H. detritum, H. marginatum, R. bursa, R. sanguineus*, and *I. ricinus* ticks (Table 1)

*Anaplasma marginale* was the most prevalent species of the genus *Anaplasma* (16/235, 6.80%), followed by *A. centrale* detected in 0.4% (1/235) of samples. The remaining 50% (17/34) of samples were confirmed by nested PCR, with the sequencing of nine positive samples revealing identity with unidentified *Anaplasma* spp. BLAST searches using the 16S rRNA gene sequence showed 99.6% identity with uncultured *Anaplasma* sp. clone AMCRO1 (Accession no. MN187218.1) in five samples, 98% with uncultured *Anaplasma* sp. clone AR2-1 (Accession no. MH250195.1) in two samples and 98% with uncultured *Anaplasma* sp. Oriente CuBov140 clone (Accession no. MK804764.1) in two samples. *A. marginale* was found in all three identified tick genera (*Rhipicephalus, Hyalomma*, and *Ixodes*), and *A. centrale* was detected only in *Rhipicephalus* (Table 1).

The genus *Borrelia* was detected in 6.8% (16/235) of the investigated ticks. None of the eight species-specific primer/probe sets used for high-throughput microfluidic PCR gave a positive signal; therefore, these samples were confirmed by nested PCR followed by sequencing. All PCR-positive samples (16/235) were confirmed by nested PCR: 10/16 were positive on gel and 4/10 were sequenced, from which only two sequences were obtained. The BLAST analysis on the *fla* gene sequence showed 92.37% identity with an unidentified *Borrelia* species (Accession no. KR677091.1).

Pathogens belonging to the genus *Bartonella* were detected in 6.8% (16/235) of the sampled ticks using the high-throughput microfluidic PCR system. Species identification was attempted on all positive samples with conventional PCR target *ftsZ* gene, but no results were obtained by sequencing.

DNA of *Babesia bigemina* was found in 6/235 (2.55%). The positive specimens of *B. bigemina* detected in *R. annulatus* and *R. sanguineus* ticks were confirmed by nested PCR and sequencing with an identity of 97.32% (Accession no. MH257721.1).

Among all the ticks analyzed, 80.4% (189/235) were positive for at least one microorganism. The level of single infection was 20% (47/235) with one microorganism; the level of co-infection was 36.6% (86/235) with two, 21.7% (51/235) with three, and 2.1% (5/235) with four microorganisms. Ticks of the genus *Ixodes* showed the highest rate of co-infection (12/13, 92.3%), followed by ticks of the genus *Hyalomma* (91/113, 80.5%) and the genus *Rhipicephalus* with a rate of co-infection of 35.7% (39/109). Double co-infection between FLE and *Rickettsia* spp. was most common in three tick’s genera identified with the respective frequencies of 48.6% (55/113), 46.2% (6/13), and 14.6% (16/109) in *Hyalomma*, *Ixodes*, and *Rhipicephalus*. Triple co-infections with FLE, *Rickettsia* spp. and *Theileria* spp. were identified with a high frequency in ticks of the genus *Ixodes* 2/13 (15.3%) followed by ticks of the genus *Hyalomma* (13/113, 11.5%) and the genus *Rhipicephalus* (3/109, 2.7%). Likewise, the highest rate of co-infection with FLE, *Rickettsia* spp. and *Anaplasma* spp. was detected in the *Ixodes* genus (3/13, 23.1%), followed by the genus *Rhipicephalus* (5/109, 4.5%) and the genus *Hyalomma* (5/113, 4.4%). Triple co-infections with FLE, and *Rickettsia* spp. either with *Borrelia* spp. or *Bartonella* spp., were observed primarily in the genus *Hyalomma* with frequencies of 5.3% (6/113). Finally, quadruple infections were found mostly in the genus *Ixodes* (1/13, 7.6%), followed by the genus *Hyalomma* (3/113, 2.6%) and the genus *Rhipicephalus* (1/109, 0.9%) (details of co-infections between different species of microorganisms are given in Appendix A).

## 3. Discussion

In this study, we identified three tick genera: *Rhipicephalus*, *Hyalomma*, and *Ixodes*. Among these, the thermophilic species *R. bursa, H. detritum*, and *H. marginatum* predominated, and the mesophilic species *I. ricinus* was less abundant (Figure 1). These results corroborate the previous reports from northern Algeria [7,11,16]. In future studies, a combination of morphological and molecular identification of ticks should be performed to identify ticks at the genus level. In North Africa, TBP detection is usually carried out using classic methods, such as PCR or real-time PCRs, which are based on the use of specific primers and/or probes [11,17,18]. This approach is limited by the characterization of a single pathogen. However, recent studies have described the importance of co-infections in the transmission of pathogens and the expression of disease severity. In this study, a new approach, based on high-throughput microfluidic technology, was used to detect 36 different microorganisms (pathogens and symbionts), and to monitor TBP circulation in hard ticks infesting cattle in northeastern Algeria.

It is important to notice that some of the ticks collected on bovines were engorged. Therefore, we cannot conclude that these ticks are vectors for the detected pathogens. The latter may have become infected with the microorganism while feeding on previously infected animals, and/or through co-feeding. Moreover, detection of DNA does not indicate that the pathogen is alive; it simply corresponds to potentially inert traces of the microorganism in the engorged tick. Here, we identified four different genera of bacteria (*Rickettsia, Anaplasma, Borrelia, Bartonella*) and two genera of protozoans (*Theileria* and *Babesia*) with an overall frequency of 80.4% of ticks infected with at least one of these TBPs. For example, *Rickettsia* spp. had the highest infection rate and four different species (i.e., *R. aeschlimannii, R. massiliae, R. monacensis*, and *R. helvetica*) were identified. Belonging to the pathogenic spotted fever group, *R. aeschlimannii* and *R. massiliae* cause infections in animals and humans worldwide [19]. The presence of *R. aeschlimannii* in *H. marginatum* ticks confirmed previous studies on the association of this vector with this bacteria species [19,20]. Surprisingly, we detected *R. aeschlimannii* DNA in *H. detritum* and *R. bursa* ticks also (Table 1). This species was first isolated from *H. marginatum* in Morocco and then in other African and Mediterranean countries [10,15,19,21,22,23]. Ticks of the genus *Hyalomma* have been reported as vectors of *R. aeschlimannii*, including *H. marginatum, H. marginatum rufipes, H. aegyptium*, and *H. truncatum* [19,23]. In addition, this bacteria species has been found in *R. appendiculatus* in South Africa and *Haemaphysalis* ticks in Spain [24], and *R. turanicus* in Greece and China [25,26]. Nevertheless, *Rickettsia* spp. is considered as an endosymbiont of *Hyalomma* ticks, and several other tick genera. *R. aeschlimannii* has been identified in a large percentage of *Hyalomma* ticks, with unknown clinical relevance [27]. According to this finding and our results, we suggest that tick species other than *Hyalomma* can also be a suitable carrier for this bacterial species.

In addition, we detected *R. massiliae* in *R. sanguineus*. This tick has been described as a vector of this species in the Mediterranean region [8], but we also amplified the DNA of this species for the first time in *Hyalomma* ticks from Algeria. A similar study in Pakistan also using the microfluidic technique amplified *R. massiliae* in *Hyalomma hussaini* and *H. anatolicum* ticks [28]. The implication of *Hyalomma* species in the transmission cycle of this pathogen needs to be clarified. In our study, *I. ricinus* harbored *R. monacensis* and *R. helvetica* that were not detected in the other tick species examined (Table 1). This confirms the previous results observed in North Africa as well as Europe [19,20,29,30,31,32].

Regarding the protozoan *Theileria* and *Babesia* species, we detected *T. orientalis* and *B. bigemina*. Although the primary vectors of *T. orientalis* are *Haemaphysalis* spp. ticks [33], in our study, *R. bursa, H. detritum, H. marginatum*, and *I. ricinus* ticks were found to be positive for this pathogen. The association of this pathogen with *R. bursa* and *R. annulatus* ticks have also been confirmed in Romania and Algeria, respectively [11,34]. Moreover, a strain closely related to *T. buffeli* has been detected in *R. sanguineus, R. bursa, R. annulatus, H. marginatum, Dermacentor marginatum*, and *Haemaphysalis punctata* ticks in Sardinia, Italy [35]. Taking these results together, the transmission of this TBP does not appear to be limited exclusively to *Haemaphysalis* spp., suggesting that other tick species may be involved in the transmission cycle worldwide. In the present study, we confirmed the presence of *B. bigemina* in Algerian *R. annulatus*, which is its principal vector [36]. Furthermore, we also found this protozoan species in *R. sanguineus*, lending support to previous reports from Sardinia and Iran [35,37]. The biological transmission of *A. marginale* involves at least 20 species of ticks mainly of the genera *Dermacentor* and *Rhipicephalus* [38]. Here, we report the presence of *A. marginale* in *R. bursa, H. detritum*, and *I. ricinus*. Similarly, *A. marginale* has been reported in *R. bursa* in Corsica and Portugal [39,40], and in *I. ricinus* in Hungary [41]. *A. centrale*, which is transmitted by *Rhipicephalus simus* [38], was confirmed individually on one *Rhipicephalus* sp. tick. *Bartonella* spp. was identified in multiple species of the tick genera *Hyalomma* and *Rhipicephalus*. There is little information on *Bartonella* transmission in cattle, but based on our recent study reporting *B. bovis* in ticks from Algeria, these ticks may play a critical role in the transmission of this pathogen in this area [42]. We also amplified DNA from *Borrelia* sp. in 16 specimens of different tick species. Usually, ticks from the genus *Ixodes* are the vectors of the zoonotic bacteria species *B. burgdorferi* s.l. from the Lyme disease group, whereas *B. theileri*, from the relapsing fever group that causes bovine borreliosis, is transmitted by *Rhipicephalus* ticks [43,44]. In previous research conducted in Algeria, DNA from these bacteria has been amplified in different tick genera in Algeria using quantitative PCR [31].

*Coxiella, Francisella*, and *Rickettsia* are the three major endosymbionts reported in ticks [6,45]. In addition to TBPs, we reported for the first time in Algeria a high rate of infection with *Francisella*-like endosymbionts in all tick species tested except *R. annulatus*. Furthermore, 60% of investigated ticks harbored unidentified *Rickettsia* spp., whereas all the specimens were negative for *Coxiella*-like endosymbionts. It has been reported that symbionts previously considered non-pathogenic may turn out to be pathogenic, as demonstrated for *R. helvetica, R. slovaca*, and *R. monacensis* [20,46]. However, the reasons that make one bacterial species become pathogenic while others remain non-pathogenic are still unclear [45]. In future studies, phylogenetic analysis targeting several genes for pathogenic and non-pathogenic microorganisms will allow us to better answer this question. Our study showed co-infection of cattle ticks with a large variety of pathogenic and non-pathogenic microorganisms. High co-infection rates of TBPs in livestock ticks has been reported in other countries [15,23,28,47]. Co-infections represent a significant risk of the cumulative effect of pathogen transfer and subsequent development of the associated diseases [48]. In addition, co-infections can cause severe complications in the treatment of tick-borne illnesses. Therefore, the study of the associations of multiple microorganisms within the same tick is of high importance, and can help better identify potential clinical co-infections to improve the epidemiological knowledge and control of TBPs [6].

## 4. Materials and Methods

### 4.1. Ethical Statement

Sample collection for this study was authorized by the National Veterinary School of Algiers, Algeria and by the Veterinary Services Department of the Tizi-Ouzou province, Algeria. All bovines were sampled according to Algerian regulations.

### 4.2. Tick Collection and Morphological Identification

A total of 1055 ticks were randomly collected from 112 bovines (with an average of nine ticks/bovine) between May 2015 and November 2017 in eight locations in the Kabylia region located in northeastern Algeria. The samples collected from each individual bovine were stored in 70% ethanol. Tick species (or genera) were determined using taxonomic keys developed by Walker et al. [49] and based on morphological characteristics observed under a stereomicroscope.

### 4.3. DNA Extraction

A random selection of one to three ticks (males and females) from each individual bovine (235 total ticks) were used for genomic DNA extraction. Prior to extraction, ticks were washed three times in sterile distilled water, dried and crushed individually using a sterile scalpel. DNA was then extracted from whole ticks using the NucleoSpin ^®^ Tissue DNA extraction kit (Macherey–Nagel, Düren, Germany) following the manufacturer’s instructions and stored at −20 ℃ until use.

### 4.4. DNA Pre-Amplification

The Perfecta PreAmp SuperMix (Quanta Biosciences, Beverly, Massachusetts, USA) was used for DNA pre-amplification according to the manufacturer’s instructions. First, all primers pair targeting TBPs were pooled, combining equal volumes with a final concentration of 0.2 μM each.

Then, reactions were carried out in a final volume of 5 μL containing 1 µL of 5× Perfecta Preamp, 1.25 μL of the pooled primer mixture, 1.5 µL of distilled water and 1.25 μL of tick DNA.

The PCR run conditions consisted of a first cycle of 95 ℃ (2 min), followed by 14 cycles of amplification at 95 ℃ (10 s) and 60 ℃ (3 min). The pre-amplified products were diluted in ultra-pure water at 1:10 and kept at −20 ℃ until use.

### 4.5. High-Throughput Microfluidic Real-Time PCR

The BioMark™ real-time PCR system (Fluidigm, San Francisco, USA) was used for the high-throughput microfluidic system, which can handle 48 real-time PCR reactions simultaneously in one single chip [6,14]. Real-time PCR reactions were performed using 6-carboxyfluorescein (FAM)- and black hole quencher (BHQ1)-labeled TaqMan probes with TaqMan Gene expression master mix in accordance with the manufacturer’s instructions (Applied Biosystems, France). Amplification consisted of 2 min at 50 ℃, 10 min at 95 ℃, followed by 40 cycles of two-step amplification of 15 s at 95 ℃, and 1 min at 60 ℃.

We carried out the high-throughput microfluidic real-time PCR to screen the bacterial and parasitic species known to circulate in ticks. We thus simultaneously targeted 36 different microorganisms belonging to 10 genera (the list of pathogens is shown in Table 2 and the list of each primer set and probes used is given in Table 3). Moreover, two primer/probe sets targeting *I. ricinus* and *R. sanguineus* tick species were used as a positive control of tick species identification and one primer/probe set targeting tick species was used to control DNA extractions (primers/probes for other tick species were not available at that time). One negative control (ultra-pure water) and one positive control DNA of *Escherichia coli* were included in each chip. The results were acquired on the BioMarkTM real-time PCR system and analyzed using the Fluidigm real-time PCR analysis software to obtain crossing point (Cp) values.

### 4.6. Standard/Nested PCR and Sequencing

Samples were considered positive for a given microorganism if the Cp value was <30, and if they were positive for a given pathogen species and its corresponding genus and negative for all other species belonging to the same genus. Other positive samples were confirmed using nested PCR or conventional PCR with primers targeting genes or regions different from those of the BioMark™ system (for the primers used, see Table 4). The PCR products were sequenced by Eurofins Genomics (https://Cochin.eurofins.com (accessed on 1 October 2020)) then assembled using BioEdit software (Ibis Biosciences, Carlsbad, CA, USA). Our results were compared in online BLAST (http://www.ncbi.nlm.nih.gov/blast (accessed on 1 October 2020)). searches against sequences publicly available in GenBank (https://www.ncbi.nlm.nih.gov/ (accessed on 1 October 2020)).

## 5. Conclusions

We used high-throughput microfluidic real-time PCR to detect TBPs in cattle ticks in Algeria. We confirmed the presence of several bacteria and protozoan species in tick-infested cattle with a fast and highly sensitive molecular method. We detected pathogenic and non-pathogenic (e.g., *Francisella*-like endosymbionts) microorganisms in ticks feeding on cattle, with a high frequency of co-infections. Further studies on endosymbionts and their possible interactions with pathogens transmitted by ticks are now needed, particularly with regard to the highly predominant ticks *Rhipicephalus* and *Hyalomma* in Algeria. Finally, our results highlight the possible involvement of tick species other than those typically reported in the transmission of some pathogens of interest in Algeria; these atypical associations deserve to be investigated further.

## Figures and Tables

**Figure 1 pathogens-10-00362-f001:**
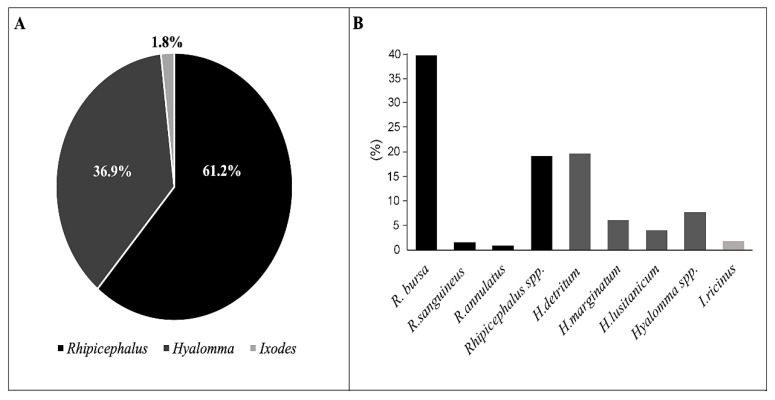
Frequencies of ixodid tick species identified in our study: (**A**) The percentages of the three genera of ticks identified. (**B**) The frequencies of specific tick species identified.

**Table 1 pathogens-10-00362-t001:** Rates of infection with tick-borne pathogens in tick species with 95% confidence intervals (CI).

Species	*Borrelia* spp.	*A.* *marginale*	*A.* *centrale*	*R.* *aeschlimannii*	*R.* *massiliae*	*R.* *monacensis*	*R.* *helvetica*	*Bartonella* spp.	*T.* *orientalis*	*B.* *bigemina*	FLE
***R. bursa*** **(*n* = 51)**	4	2	0	11	0	0	0	3	8	0	15
	(7.8%)	(3.9%)		(21.5%)				(5.8%)	(15.6%)		(29.4%)
	(0.4–15.2%)	(0–9.2%)		(10.2–32.5%)				(0–12.2%)	(5.7–25.6%)		(16.8–41.9%)
***R. sanguineus*** **(*n* = 07)**	1	0	0	0	4	0	0	2	0	5	1
	(14.2%)				(57.1%)			(28.5%)		(71.4%)	(14.2%)
	(0–40%)				(20.4–93.8%)			(0–61.9%)		(37.9–100%)	(0–40.2%)
***R. annulatus*** **(*n* = 01)**	0	0	0	0	0	0	0	0	0	1	0
										(100%)	
										(0–100%)	
***Rhipicephalus* spp.** **(*n* = 50)**	1	6	1	5	0	0	0	3	11	0	21
	(2%)	(12%)	(2%)	(10%)				(6%)	(22%)		(42%)
	(0–5.8%)	(3–21%)	(0–5.8%)	(1.6–18.3%)				(0–12.5%)	(10.5–33.4%)		(28.3–55.6%)
***H. detritum*** **(*n* = 41)**	3	2	0	2	2	0	0	3	6	0	37
	(7.3%)	(4.8%)		(4.8%)	(4.8%)			(7.3%)	(14.3%)		(90.2%)
	(0–15.2%)	(0–11.3%)		(0–11.3%)	(0–11.3%)			(0–15.2%)	(3.5–25%)		(81.1–99.3%)
***H. marginatum*** **(*n* = 15)**	0	0	0	1	0	0	0	0	1	0	12
				(6.6%)					(6.6%)		(80%)
				(0–19.1%)					(0–19.1%)		(59.7–100%)
***H. lusitanicum*** **(*n* = 04)**	0	0	0	0	0	0	0	1	0	0	4
								(25%)			(100%)
								(0–67.4%)			(25–100%)
***Hyalomma* spp.** **(*n* = 53)**	6	1	0	8	5	0	0	4	10	0	47
	(11.3%)	(1.8%)		(15.1%)	(9.4%)			(7.5%)	(18.8%)		(88.6%)
	(2.7–19.8%)	(0–5.3%)		(5.9–25.6%)	(1.5–17.2%)			(0.2–14.1%)	(8.2–29.3%)		(80–97.1%)
***I. ricinus*** **(*n* = 13)**	1	5	0	0	0	5	2	0	6	0	11
	(7.6%)	(38.4%)				(38.4%)	(15.3%)		(46.1%)		(84.6%)
	(0–22%)	(12–64.8%)				(12–64.8%)	(0–34.8%)		(19–73.1%)		(64.9–100%)
**Total** **(*n* = 235)**	**16**	**16**	**1**	**27**	**11**	**5**	**2**	**16**	**42**	**6**	**148**
	**6.8%**	**6.8%**	**0.4%**	**11.5%**	**4.6%**	**2.1%**	**0.8%**	**6.8%**	**17.8%**	**2.5%**	**62.9%**
	**(3.5–10%)**	**(3.5–10%)**	**(0–1.2%)**	**(7.4–15.5%)**	**(1.9–7.2%)**	**(0.2–3.9%)**	**(0–1.9%)**	**(3.5–10%)**	**(13–22.6%)**	**(0.5–4.4%)**	**(56.7–69%)**

FLE: *Francisella*-like endosymbiont.

**Table 2 pathogens-10-00362-t002:** Bacteria and parasites targeted in our study.

	Genus	Species	Numbers
**Bacteria**	*Borrelia*	*B. burgdorferi senso stricto, B. garinii, B. afzelii, B. valaisiana, B. lusitaniae, B. spielmanii, B. bissettii, B. miyamotoi.*	8
	*Anaplasma*	*A. marginale, A. platys, A. phagocytophilum, A.ovis, A. centrale, A. bovis.*	6
	*Ehrlichia*	*E. ruminantium, Neoehrlichia mikurensis.*	2
	*Rickettsia*	*R. conorii, R. slovaca, R. massiliae, R. prowazekii, R. aeschlimannii, R. andeanae, R. typhi, R. akari*	8
	*Bartonella*	*B. henselae*	1
	*Francisella*	*F. tularensis, Francisella*-like endosymbionts.	2
	*Coxiella*	*C. burnettii.*	1
**Parasites**	*Babesia*	*B. microti, B. ovis, B. bigemina, B. bovis, B. caballi, B. divergens.*	6
	*Theileria*	*T. mutans, T. velifera.*	2
	*Hepatozoon*	*Hepatozoon* spp.	
**Total**	**10**		**36**

**Table 3 pathogens-10-00362-t003:** List of primers and probes used in this study for microfluidic real-time PCR [6,14,47].

Pathogen	Target Gene	Primers (F, R; 5′-3′) and Probe (P)	Length (bp)
***Borrelia burgdorferi s.s.***	*rpoB*	F-GCTTACTCACAAAAGGCGTCTT	83
		R-GCACATCTCTTACTTCAAATCCT	
		P-AATGCTCTTGGACCAGGAGGACTTTCA	
***Borrelia garinii***	*rpoB*	F-TGGCCGAACTTACCCACAAAA	88
		R-ACATCTCTTACTTCAAATCCTGC	
		P-TCTATCTCTTGAAAGTCCCCCTGGTCC	
***Borrelia afzelii***	*fla*	F-GGAGCAAATCAAGATGAAGCAAT	116
		R-TGAGCACCCTCTTGAACAGG	
		P-TGCAGCCTGAGCAGCTTGAGCTCC	
***Borrelia valaisiana***	*ospA*	F-ACTCACAAATGACAGATGCTGAA	135
		R-GCTTGCTTAAAGTAACAGTACCT	
		P-TCCGCCTACAAGATTTCCTGGAAGCTT	
***Borrelia lusitaniae***	*rpoB*	F-CGAACTTACTCATAAAAGGCGTC	87
		R-TGGACGTCTCTTACTTCAAATCC	
		P-TTAATGCTCTCGGGCCTGGGGGACT	
***Borrelia spielmanii***	*fla*	F-ATCTATTTTCTGGTGAGGGAGC	71
		R-TCCTTCTTGTTGAGCACCTTC	
		P-TTGAACAGGCGCAGTCTGAGCAGCTT	
***Borrelia bissettii***	*rpoB*	F-GCAACCAGTCAGCTTTCACAG	118
		R-CAAATCCTGCCCTATCCCTTG	
		P-AAAGTCCTCCCGGCCCAAGAGCATTAA	
***Borrelia miyamotoi***	*glpQ*	F-CACGACCCAGAAATTGACACA	94
		R-GTGTGAAGTCAGTGGCGTAAT	
		P-TCGTCCGTTTTCTCTAGCTCGATTGGG	
***Borrelia*** **spp.**	*23S rRNA*	F-GAGTCTTAAAAGGGCGATTTAGT	73
		R-CTTCAGCCTGGCCATAAATAG	
		P-AGATGTGGTAGACCCGAAGCCGAGT	
***Anaplasma marginale***	*msp1*	F-CAGGCTTCAAGCGTACAGTG	85
		R-GATATCTGTGCCTGGCCTTC	
		P-ATGAAAGCCTGGAGATGTTAGACCGAG	
***Anaplasma platys***	*groEL*	F-TTCTGCCGATCCTTGAAAACG	75
		R-CTTCTCCTTCTACATCCTCAG	
		P-TTGCTAGATCCGGCAGGCCTCTGC	
***Anaplasma phagocytophilum***	*msp2*	F-GCTATGGAAGGCAGTGTTGG	77
		R-GTCTTGAAGCGCTCGTAACC	
		P-AATCTCAAGCTCAACCCTGGCACCAC	
***Anaplasma ovis***	*msp4*	F-TCATTCGACATGCGTGAGTCA	92
		R-TTTGCTGGCGCACTCACATC	
		P-AGCAGAGAGACCTCGTATGTTAGAGGC	
***Anaplasma centrale***	*groEL*	F-AGCTGCCCTGCTATACACG	79
		R-GATGTTGATGCCCAATTGCTC	
		P-CTTGCATCTCTAGACGAGGTAAAGGGG	
***Anaplasma bovis***	*groEL*	F-GGGAGATAGTACACATCCTTG	73
		R-CTGATAGCTACAGTTAAGCCC	
		P-AGGTGCTGTTGGATGTACTGCTGGACC	
***Anaplasma*** **spp.**	*16S rRNA*	F-CTTAGGGTTGTAAAACTCTTTCAG	160
		R-CTTTAACTTACCAAACCGCCTAC	
		P-ATGCCCTTTACGCCCAATAATTCCGAACA	
***Ehrlichia*** **spp.**	*16S rRNA*	F-GCAACGCGAAAAACCTTACCA	98
		R-AGCCATGCAGCACCTGTGT	
		P-AAGGTCCAGCCAAACTGACTCTTCCG	
***Ehrlichia ruminantium***	*gltA*	F-CCAGAAAACTGATGGTGAGTTAG	116
		R-AGCCTACATCAGCTTGAATGAAG	
		P-AGTGTAAACTTGCTGTTGCTAAGGTAGCATG	
***Neoehrlichia mikurensis***	*groEL*	F-AGAGACATCATTCGCATTTTGGA	96
		R-TTCCGGTGTACCATAAGGCTT	
		P-AGATGCTGTTGGATGTACTGCTGGACC	
***Rickettsia conorii***	*23S-5S ITS*	F-CTCACAAAGTTATCAGGTTAAATAG	118
		R-CGATACTCAGCAAAATAATTCTCG	
		P-CTGGATATCGTGGCAGGGCTACAGTAT	
***Rickettsia slovaca***	*23S-5S ITS*	F-GTATCTACTCACAAAGTTATCAGG	138
		R-CTTAACTTTTACTACAATACTCAGC	
		P-TAATTTTCGCTGGATATCGTGGCAGGG	
***Rickettsia massiliae***	*23S-5S ITS*	F-GTTATTGCATCACTAATGTTATACTG	128
		R-GTTAATGTTGTTGCACGACTCAA	
		P-TAGCCCCGCCACGATATCTAGCAAAAA	
***Rickettsia prowazekii***	*gltA*	F-CAAGTATCGGTAAAGATGTAATCG	151
		R-TATCCTCGATACCATAATATGCC	
		P-ATATAAGTAGGGTATCTGCGGAAGCCGAT	
***Rickettsia aeschlimannii***	*ITS*	F-CTCACAAAGTTATCAGGTTAAATAG	134
		R-CTTAACTTTTACTACGATACTTAGCA	
		P-TAATTTTTGCTGGATATCGTGGCGGGG	
***Rickettsia*** ***andeanae***	*OmpB*	F-GGCGGACAGGTAACTTTTGG	165
		R-AAGGATCATAGTATCAGGAACTG	
		P- ACACATAGTTGACGTTGGTACAGACGGTAC	
***Rickettsia*** ***typhi***	*OmpB*	F-CAGGTCATGGTATTACTGCTCA	133
		R-GCAGCAGTAAAGTCTATTGATCC	
		P-ACAAGCTGCTACTACAAAAAGTGCTCAAAATG	
***Rickettsia*** ***akari***	*OmpB*	F-GTGCTGTTGCAGGTGGTAC	101
		R-TAAAGTAATACCGTGTAATGCAGC	
		P-ATTACCAGCACCGTTACCTATATCACCGG	
***Rickettsia*** **spp.**	*gltA*	F-GTCGCAAATGTTCACGGTACTT	78
		R-TCTTCGTGCATTTCTTTCCATTG	
		P-TGCAATAGCAAGAACCGTAGGCTGGATG	
***Bartonella henselae***	*pap31*	F-CCGCTGATCGCATTATGCCT	107
		R-AGCGATTTCTGCATCATCTGCT	
		P-ATGTTGCTGGTGGTGTTTCCTATGCAC	
***Bartonella*** **spp.**	*ssrA*	F-CGTTATCGGGCTAAATGAGTAG	118
		R-ACCCCGCTTAAACCTGCGA	
		P-TTGCAAATGACAACTATGCGGAAGCACGTC	
***Francisella tularensis***	*tul4*	F-ACCCACAAGGAAGTGTAAGATTA	76
		R-GTAATTGGGAAGCTTGTATCATG	
		P-AATGGCAGGCTCCAGAAGGTTCTAAGT	
***Francisella*** **-like endosymbionts**	*fopA*	F-GGCAAATCTAGCAGGTCAAGC	91
		R-CAACACTTGCTTGAACATTTCTAG	
		P-AACAGGTGCTTGGGATGTGGGTGGTG	
***Coxiella burnettii***	*IS1111*	F-TGGAGGAGCGAACCATTGGT	86
		R-CATACGGTTTGACGTGCTGC	
		P-ATCGGACGTTTATGGGGATGGGTATCC	
***Coxiella burnettii***	*idc*	F-AGGCCCGTCCGTTATTTTACG	74
		R-CGGAAAATCACCATATTCACCTT	
		P-TTCAGGCGTTTTGACCGGGCTTGGC	
***Babesia microti***	*CCTeta*	F-ACAATGGATTTTCCCCAGCAAAA	145
		R-GCGACATTTCGGCAACTTATATA	
		P-TACTCTGGTGCAATGAGCGTATGGGTA	
***Babesia ovis***	*18SrRNA*	F-TCTGTGATGCCCTTAGATGTC	92
		R-GCTGGTTACCCGCGCCTT	
		P-TCGGAGCGGGGTCAACTCGATGCAT	
***Babesia bigemina***	*18SrRNA*	F-ATTCCGTTAACGAACGAGACC	99
		R-TTCCCCCACGCTTGAAGCA	
		P-CAGGAGTCCCTCTAAGAAGCAAACGAG	
***Babesia bovis***	*CCTeta*	F-GCCAAGTAGTGGTAGACTGTA	100
		R-GCTCCGTCATTGGTTATGGTA	
		P-TAAAGACAACACTGGGTCCGCGTGG	
***Babesia caballi***	*Rap1*	F-GTTGTTCGGCTGGGGCATC	94
		R-CAGGCGACTGACGCTGTGT	
		P-TCTGTCCCGATGTCAAGGGGCAGGT	
***Babesia divergens***	*hsp70*	F-CTCATTGGTGACGCCGCTA	83
		R-CTCCTCCCGATAAGCCTCTT	
		P-AGAACCAGGAGGCCCGTAACCCAGA	
***Theileria mutans***	*ITS*	F-CCTTATTAGGGGCTACCGTG	119
		R-GTTTCAAATTTGAAGTAACCAAGTG	
		P-ATCCGTGAAAAACGTGCCAAACTGGTTAC	
***Theileria velifera***	*18S rRNA*	F-TGTGGCTTATCTGGGTTCGC	151
		R-CCATTACTTTGGTACCTAAAACC	
		P-TTGCGTTCCCGGTGTTTTACTTTGAGAAAG	
***Theileria*** **spp.**	*18S*	F-TGAACGAGGAATGCCTAGTATG	104
		R-CACCGGATCACTCGATCGG	
		P-TAGGAGCGACGGGCGGTGTGTAC	
***Hepatozoon*** **spp.**	*18S rRNA*	F-ATTGGCTTACCGTGGCAGTG	175
		R-AAAGCATTTTAACTGCCTTGTATTG	
		P-ACGGTTAACGGGGGATTAGGGTTCGAT	
**Tick species**	*16SrRNA*	F-AAATACTCTAGGGATAACAGCGT	99
		R-TCTTCATCAAACAAGTATCCTAATC	
		P-CAACATCGAGGTCGCAAACCATTTTGTCTA	
***Rhipicephalus*** ***sanguineus***	*ITS2*	F-TTGAACGCTACGGCAAAGCG	110
		R-CCATCACCTCGGTGCAGTC	
		P-ACAAGGGCCGCTCGAAAGGCGAGA	
***Ixodes ricinus***	*ITS2*	F-CGAAACTCGATGGAGACCTG	77
		R-ATCTCCAACGCACCGACGT	
		P-TTGTGGAAATCCCGTCGCACGTTGAAC	
***Escherichia coli***	*eae*	F-CATTGATCAGGATTTTTCTGGTGATA	102
		R-CTCATGCGGAAATAGCCGTTA	
		P-ATAGTCTCGCCAGTATTCGCCACCAATACC	

F: forward; R: reverse; P: probe; bp: base pairs.

**Table 4 pathogens-10-00362-t004:** List of primers used in this study for confirmation using nested and conventional PCR.

Pathogen	Target Gene	Primer Name	Sequence (5′-3′)	Amplicon Size (bp)	T	Reference
***Borrelia*** **spp.**	*FlaB*	FlaB280 F	GCAGTTCARTCAGGTAACGG	645	55	[50]
		FlaL R	GCAATCATAGCCATTGCAGATTGT			
		FlaB_737F	GCATCAACTGTRGTTGTAACATTAACAGG			
		FlaLL R	ACATATTCAGATGCAGACAGAGGT	407		
***Anaplasma*** **spp.**	*16S rRNA*	EHR1 F	GAACGAACGCTGGCGGCAAGC	693	60	[51]
		EHR2 R	AGTA(T/C)CG(A/G)ACCAGATAGCCGC			
		EHR3 F	TGCATAGGAATCTACCTAGTAG			
		EHR2 R	AGTA(T/C)CG(A/G)ACCAGATAGCCGC	629	55	
***Rickettsia*** **spp.**	*gltA*	Rsfg877	GGG GGC CTG CTC ACG GCG G	381	56	[52]
		Rsfg1258	ATT GCA AAA AGT ACA GTG AAC A			
***Bartonella*** **spp.**	*ftsZ*	257 F	GCCTTCAAGGAGTTGATTTTGTTGTTGCCA	580	55	[53]
		258 R	ACGACCCATTTCATGCATAACAGAAC			
***Babesia/*** ***Theileria*** ***/Hepatozoon*** **spp.**	*18S rRNA*	BTH 18S 1st F	GTGAAACTGCGAATGGCTCATTAC	1500	58	[54]
		BTH 18S 1st R	AAGTGATAAGGTTCACAAAACTTCCC			
		BTH 18S 2nd F	GGCTCATTACAACAGTTATAGTTTATTTG			
		BTH 18S 2nd R	CGGTCCGAATAATTCACCGGAT			

F: forward; R: reverse; bp: base pairs; T: hybridization temperature.

## Data Availability

Data are available under request to corresponding author s and published in GenBank for sequences.

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
