# Peer review of "High-Throughput Microfluidic Real-Time PCR for the Detection of Multiple Microorganisms in Ixodid Cattle Ticks in Northeast Algeria"

_pathogens, 2021, doi:10.3390/pathogens10030362_

Round 1

Reviewer 1 Report

The review in the file below.

Author Response

The article presented for review is a very carefully and well-written study of extremely

important research results. The methods used are modern and well-chosen.

Answer : We would like to thanks the reviewer for his comments

Nevertheless, I have listed my little comments and questions below:

Lines 75-80, generic and species names should be written in italics.

Answer : Corrections have been made

Table 1, Confidence intervals (95% CI) should be included.

Answer : 95% CI have been added

It would also be good to determine whether the level of infection between different species of ticks was statistically significant (e.g. Chi square).

Answer: Because the size of the different tick population were too different, we could not perform statistical analysis (even Chi Square) with enough confident in the results. So we prefer not add this information.

Besides, it seems to me that the answers to the following questions would complete the chapter

“Materials and methods”:

  1. Were only female ticks analyzed?

Answer : Male and females were analysed. No nymphs or larvae were collected. This information was added in the revised version

  1. Was DNA isolated from whole ticks, or were fully engorged ticks bisected?

Answer: DNA was extracted from whole ticks

  1. Couldn't Borrelia species be identified by sequencing and sequence analysis?

Answer: As mentioned into the result section Borrelia spp was confirmed by sequencing and blast analysis identified it as an unidentified Borrelia “The genus Borrelia was detected in 6.8% (16/235) of the investigated ticks. None of the eight species-specific primer/probe sets used for high-throughput microfluidic PCR gave a positive signal; therefore, these samples were confirmed by nested PCR followed by sequencing. All PCR-positive samples (16/235) were confirmed by nested PCR: 10/16 were positive on gel and 4/10 were sequenced, from which only two sequences were obtained. The BLAST analysis on the fla gene sequence showed 92.37% identity with an unidentified Borrelia species (Accession no. KR677091.1). »

However, I would like to emphasize once again that the above comments and questions do not affect my very high rating of the manuscript.

Answer: Thanks again for your comments

Reviewer 2 Report

This manuscript describes identification of tick-borne pathogens (TBP) in ticks infesting cattle in Algeria. The investigation of microorganism in ticks is important both for the understanding of the circulation and epidemiology of tick-borne diseases, the composition of tick microbiome and the relationship between pathogenic and non-pathogenic species within the tick environment. The manuscript is clearly written, fits the scope of the journal and warrants publication after some adjustments.

General comments:

  • The main shortcoming of the methodology in this study was the incomplete identification of the tick species. Since morphological identification is often unclear and requires much skill, it is possible to use molecular methods to distinguish between species of the same genus. Furthermore, when selecting the ticks for pathogen screening, it would have been beneficial to select well-defined samples rather than ticks classified only to the genus level. If possible, I suggest the use of molecular methods to identify, at least, the selected ticks. (If not possible- please address this issue in the discussion).
  • When surveying the infection of individual ticks co-infection with multiple species was briefly mentioned (and only specified in the supplement). The subject of co-infection in ticks have been addressed in multiple studies in the past and some suggested that co-infection occurs more often than expected. I think a summary of your results of co-infection in individuals ticks is warranted, especially if there is association between specific microorganisms in the different tick species/genera.
  • The analysis of all cattle-ticks together is problematic, since both vector competence and microbiome differ between tick species. If the identification of ticks was done only to the genus level, than, at least, refer to pathogen carriage in each genus.
  • When interpreting the results, it is important to consider that detection of microorganisms in a tick does not necessarily imply vector competence. Pathogen DNA may originate from the tick bloodmeal, without completion of its life cycle. Some microorganisms may survive in the tick environment without being transmitted in the bloodmeal (with or without being considered as endosymbionts). These issues should be better addressed in the discussion.

Specific comments:

  • Line 38 – protozoa
  • Line 39 – Do you mean “the mammal host”?
  • Lines 60-62 – In recent years there are numerous studies investigating the microbiome of various tick species and the interactions between the microbiome and transmission of pathogens.
  • Line 74 – were immature stages not found, or not selected during sampling? R. annulatus is one-host species, and immature stages are expected.
  • Line 80 – In the future, I suggest using molecular tools for classification, to classify to the species level without relying on morphological tools that require much skill.
  • Line 95 and thereafter – there were 186 Rickettsia-positive ticks (and not 235). In addition, the classification of R. spp. does not reach 100% (nor 186 ticks).
  • Line 97 – were these selected from the “unidentified” samples? Was the gltA gene specific enough? (was the 100% homology only to one R. species? Was it confirmed with an additional gene?)
  • Line 106 , and later 161-186– R. spp. is considered as an endosymbiont of Hyalomma ticks, several other tick genera. Specifically R. aeschlimannii, have been identified in a large percentage of hyalomma ticks (as the authors stated in the discussion), with unknown clinical relevance.

Reviered in: [Socolovschi, C., Mediannikov, O., Raoult, D. and Parola, P., 2009. The relationship between spotted fever group Rickettsiae and ixodid ticks. Veterinary research40(2), pp.1-20.]

With numerous other papers reporting its presence in various tick species.

  • Line 143 – Table 1 – Since the number of ticks appears on the left, it is unnecessary to repeat it in each column. Perhaps you can replace it with % positive next to the number.
  • Lines 215-230 – As the authors state, the microbiome of ticks sometimes consists of closely related pathogenic and non-pathogenic species. Therefore, identification of potentially pathogenic organisms within ticks does not necessarily indicate vector competence for the transmission of these microorganisms, especially from the uppermentioned species (mostly rickettsia in this case). Therefore, the interpretation of the results should be made with caution.
  • Line 284 – I understand that the authors considered the possibility of cross-positivity between closely related species and therefore, required a unique reaction. However, co-infection with multiple species is possible, and should be confirmed by sequencing of one or more unique target genes (as the authors have done to confirm single infection).

Author Response

The main shortcoming of the methodology in this study was the incomplete identification of the tick species. Since morphological identification is often unclear and requires much skill, it is possible to use molecular methods to distinguish between species of the same genus. Furthermore, when selecting the ticks for pathogen screening, it would have been beneficial to select well-defined samples rather than ticks classified only to the genus level. If possible, I suggest the use of molecular methods to identify, at least, the selected ticks. (If not possible- please address this issue in the discussion).

Answer: We thanks the reviewer for this valuable comment. Morphological identification was performed in Algeria with researchers specialized in Tick identification. And our result corroborate with previous reports from Algeria [16,7,11] (as mentioned in the discussion). Then DNA extraction and Pathogens/Symbionts detection were performed in France. Unfortunately we didn’t use molecular tool to confirm our morphological identification due to volume of DNA available after the several confirmation of pathogens/symbionts by Nested PCR. We agree this important point and discuss it into the discussion section (one sentence added).

When surveying the infection of individual ticks co-infection with multiple species was briefly mentioned (and only specified in the supplement). The subject of co-infection in ticks have been addressed in multiple studies in the past and some suggested that co-infection occurs more often than expected. I think a summary of your results of co-infection in individuals ticks is warranted, especially if there is association between specific microorganisms in the different tick species/genera.

The analysis of all cattle-ticks together is problematic, since both vector competence and microbiome differ between tick species. If the identification of ticks was done only to the genus level, than, at least, refer to pathogen carriage in each genus.

Answer: In the table and result section, single infection rates by each pathogens/symbionts were detailed for each species/genera of ticks. But we agreed with reviewer comment regarding the missing information for co-infection and added more information into the result section.

The following paragraph has been added: “« Ticks of the genus Ixodes have the highest rate of co-infection (12/13, 92.3%), followed by ticks of the genus Hyalomma (91/113, 80.5%) and finally the genus Rhipicephalus with a rate of co-infection of 35.7% (39/109). Double co-infection between FLE and Rickettsia spp. was most common in the 3 tick’s genera identified with the respective frequencies of 48.6% (55/113), 46.% (6/13) and 14.6% (16/109) in Hyalomma, Ixodes and Rhipicephalus. Triple co-infections with FLE + Rickettsia spp. + Theileria spp. were identified with a high frequency in ticks of the genus Ixodes 2/13 (15.3%) followed by ticks of the genus Hyalomma (13/113, 11.5%) and finally the genus Ripicephalus (3/109, 2.7%). Likewise for “FLE + Rickettsia spp. + Anaplasma spp.” , the genus Ixodes has the highest triple-infection rate (3/13, 23%), followed by the genus Ripicephalus (5/109, 4.5%) and finally the genus Hyalomma (5/113, 4.4%). Regarding triple co-infections with (FLE + Rickettsia spp. + Borrelia spp.) and (FLE + Rickettsia spp. + Bartonella spp.), They were observed mainly in the genus Hyalomma with frequencies of 5.3% (6/113) respectively. Finally, tetra-infections were found mostly in 1/13 (7.6%) of the genus Ixodes, 3/113 (2.6%) of the genus Hyalomma and 1/109 (0.9%) of the genus Rhipicephalus (Table S1)”

When interpreting the results, it is important to consider that detection of microorganisms in a tick does not necessarily imply vector competence. Pathogen DNA may originate from the tick bloodmeal, without completion of its life cycle. Some microorganisms may survive in the tick environment without being transmitted in the bloodmeal (with or without being considered as endosymbionts). These issues should be better addressed in the discussion.

Answer: We fully agree with reviewer comment that’s why this statement was already mentioned into the original version of the manuscript “However, some of the ticks collected on bovines were engorged. Therefore, we cannot conclude that these ticks are vectors for the detected pathogens. The latter may have become infected with the microorganism while feeding on previously infected animals, and/or through co-feeding. Moreover, detection of DNA does not indicate that the pathogen is alive; it simply corresponds to potentially inert traces of the microorganism in the engorged tick.” To be more clear, this sentence has been put at the beginning of the paragraph.

Specific comments:

Line 38 – protozoa

Answer: word has been modified accordingly.

Line 39 – Do you mean “the mammal host”?

Answer: The words has been modified accordingly.

Lines 60-62 – In recent years there are numerous studies investigating the microbiome of various tick species and the interactions between the microbiome and transmission of pathogens.

Answer: We agree with reviewer comments, several studies dealt with microbiome of ticks and their interaction with pathogens, nevertheless only few studies were performed in the studied area. The sentence has been modified.

Line 74 – were immature stages not found, or not selected during sampling? R. annulatus is one-host species, and immature stages are expected.

Answer: We didn’t observed any immature stages during tick collection even if we collected all the ticks founded on the animal. For R. annulatus ticks, we collected 9 specimens, all were females and adults, following the taxonomic identification keys used.

Line 80 – In the future, I suggest using molecular tools for classification, to classify to the species level without relying on morphological tools that require much skill.

Answer: We fully agreed with this suggestion from the reviewer, and added a sentence in the discussion section as proposed by reviewer 1 and 2.

Line 95 and thereafter – there were 186 Rickettsia-positive ticks (and not 235).

Answer: Yes, that is true, the number of Rickttsia-positive ticks is equal to 186. We have chosen to divide over 235 ticks (the total of ticks analyzed) in order to calculate the Prevalence/Infection rate for each pathogen/symbiont detected in each tick species/genus. We changed “Among the Rickettsia-positive ticks » by “Among the Rickettsia-tested ticks”.

In addition, the classification of R. spp. does not reach 100% (nor 186 ticks).

Answer: We are sorry but following the results mentioned below the result is 186. 27+11+62+86= 186

R. aeschlimannii

R. massiliae

undetermined Rickettsia spp.

multiple Rickettsia species at the same time

Total

27

11

62

86

186

Line 97 – were these selected from the “unidentified” samples? Was the gltA gene specific enough? (was the 100% homology only to one R. species? Was it confirmed with an additional gene?)

Answer: For ticks infected with one Rickettsia (single infection), 10 Rickettsia specimens were chosen randomly and sequenced from the different Rickettsia positive samples (unidentified samples and identified species to confirm our results). No other gene than gltA was tested.

Line 106 , and later 161-186– R. spp. is considered as an endosymbiont of Hyalomma ticks, several other tick genera. Specifically R. aeschlimannii, have been identified in a large percentage of hyalomma ticks (as the authors stated in the discussion), with unknown clinical relevance. Reviered in: [Socolovschi, C., Mediannikov, O., Raoult, D. and Parola, P., 2009. The relationship between spotted fever group Rickettsiae and ixodid ticks. Veterinary research, 40(2), pp.1-20.]With numerous other papers reporting its presence in various tick species.

Answer: We agreed with reviewer comments and added this information and modified our sentence into the discussion section. “Nevertheless, Rickettsia spp. is considered as an endosymbiont of Hyalomma ticks, and several other tick genera. Specifically R. aeschlimannii, have been identified in a large percentage of Hyalomma ticks, with unknown clinical relevance (Socolovschi et al. 2009). According to this finding and our results, we suggest that tick species other than Hyalomma can also be suitable carrier for this bacterial species. »

Line 143 – Table 1 – Since the number of ticks appears on the left, it is unnecessary to repeat it in each column. Perhaps you can replace it with % positive next to the number.

Answer: Table 1 has been modified as suggested.

Lines 215-230 – As the authors state, the microbiome of ticks sometimes consists of closely related pathogenic and non-pathogenic species. Therefore, identification of potentially pathogenic organisms within ticks does not necessarily indicate vector competence for the transmission of these microorganisms, especially from the uppermentioned species (mostly rickettsia in this case). Therefore, the interpretation of the results should be made with caution.

Answer: We fully agree with reviewer comment that’s why this statement was already mentioned into the original version of the manuscript “However, some of the ticks collected on bovines were engorged. Therefore, we cannot conclude that these ticks are vectors for the detected pathogens. The latter may have become infected with the microorganism while feeding on previously infected animals, and/or through co-feeding. Moreover, detection of DNA does not indicate that the pathogen is alive; it simply corresponds to potentially inert traces of the microorganism in the engorged tick.” To be more clear, this sentence has been put at the beginning of the paragraph.

Line 284 – I understand that the authors considered the possibility of cross-positivity between closely related species and therefore, required a unique reaction. However, co-infection with multiple species is possible, and should be confirmed by sequencing of one or more unique target genes (as the authors have done to confirm single infection).

Answer: According to our results, we found positive samples for both multiple species of the same genus only for Rickettsia. And since co-infections between Rickettsia species are rarely described in the literature, we sequenced 10 specimens with possible co-infection (10/86) and the BLAST results for gltA gene indicated 100% identity with an uncultured Rickettsia sp. (Accession no. KU596570.1). Unfortunately this confirmation was done with only one gene gltA.

Reviewer 3 Report

The study done by Boularias et al., showed the presence of several bacteria and protozoan species in tick-infested cattle using high-throughput microfluidic real-time PCR. The authors reported that the prevalence of tick-borne microorganisms in ixodid cattle ticks in northeast Algeria as the following: Rickettsia (79.1%),  Francisella-like endosymbionts (62.9%), Theileria (17.8%), Anaplasma  (14.4%), Bartonella spp. (6.8%), Borrelia spp. (6.8%) and Babesia spp. (2.5%). 

Recent reports discuss this point before in the same region in Algeria (PMID: 31786146). The novelty point is the use of microfluidic real-time PCR. However, the authors need to address the following points

a) What is the difference between this assay and previously published assays such as nested PCR regarding the sensitivity? Or in the other meaning, what are advantages of this methodology compared to other known assays?

b) The authors develop this technique from ticks. Did the authors try the same methodology using animal samples?

c)The authors did sequencing and blast for the isolated microorganisms. It is better for the author to do phylogenetic tree and to show the degree of relatedness especially with the isolates recently detected in the same region in Algeria .

d) The probe used in this assay as mentioned by the authors in the methodology used one fluorophore (FAM) and quencher (Black hole). Can the authors explain if this assay is suitable for detection of coinfection? and how can the analysis be done if the same  fluorophore  and quencher are used? Please explain in details in methodology.

Author Response

The study done by Boularias et al., showed the presence of several bacteria and protozoan species in tick-infested cattle using high-throughput microfluidic real-time PCR. The authors reported that the prevalence of tick-borne microorganisms in ixodid cattle ticks in northeast Algeria as the following: Rickettsia (79.1%), Francisella-like endosymbionts (62.9%), Theileria (17.8%), Anaplasma (14.4%), Bartonella spp. (6.8%), Borrelia spp. (6.8%) and Babesia spp. (2.5%). Recent reports discuss this point before in the same region in Algeria (PMID: 31786146). The novelty point is the use of microfluidic real-time PCR. However, the authors need to address the following points

a) What is the difference between this assay and previously published assays such as nested PCR regarding the sensitivity? Or in the other meaning, what are advantages of this methodology compared to other known assays?

Answer: This methodology was developed by the team of Sara Moutailler (see Michelet et al., 2014 for details) and allow the screening of 48 or 96 samples (ticks or animals) against 48 or 96 targets (pathogens or symbionts) in one experiment (2304 or 9216 reactions simultaneously), using realtime microfluidic PCR. Each sample will meet each target into an individual chamber (microfluidic chamber), so it is not multiplex realtime PCR but highthroughput individual realtime PCR (FAM, BHQ1). During the development of the technic, specificity and sensitivity of each primers/probe set has been tested for each targeted microorganism. This technic allow a high level of sensitivity / identical to realtime PCR but using a very small volume of DNA (1.5µL used). To allow a highest sensitivity, a preamplification step was also added prior to microfluidic realtime PCR allowing to increase the detection of microorganism DNAs inside the total DNA extracted from each tick/sample. This technic has been used in several epidemiological studies on tick-borne pathogens in different region of the world (see author Moutailler S) and was also used by colleagues in Danemark (see author Bodker R).

b) The authors develop this technique from ticks. Did the authors try the same methodology using animal samples?

Answer: This method was first developed for ticks and was then used in different kind of samples with success: Ticks (several international publications) ; Animals (blood and organs ; Malmsten et al 2018 ; Vector Borne and Zoonotic Diseases. 10.1089/vbz.2018.2277) and humans (blood ; publication under preparation). This methodology was also developed to detect tick-borne viruses (Gondard et al., 2018) and mosquito-borne viruses (Moutailler et al., 2019).

c)The authors did sequencing and blast for the isolated microorganisms. It is better for the author to do phylogenetic tree and to show the degree of relatedness especially with the isolates recently detected in the same region in Algeria .

Answer: We agreed with reviewer comments but unfortunately, we didn’t performed confirmation and sequencing with several targeted genes (only one gene used for each microorganism), this will not allow us to perform informative phylogenetic trees. Moreover, the objective of our study was to identify the main microorganisms (single and coinfection) in ticks from Algeria using a fast and high sensitive method. This surveillance method represents a major improvement in epidemiological studies, able to facilitate comprehensive testing of TBPs, and which can also be customized to monitor emerging diseases.

d) The probe used in this assay as mentioned by the authors in the methodology used one fluorophore (FAM) and quencher (Black hole). Can the authors explain if this assay is suitable for detection of coinfection? and how can the analysis be done if the same fluorophore and quencher are used? Please explain in details in methodology.

Answer: Each primers/probe set was a FAM/BHQ1. Each primers/probe set target a specific species of microorganism or a genus of microorganism. Then each sample (48 or 96 samples) is tested for each microorganism (48 or 96 targeted microorganism) in individual microfluidic chamber (total volume reaction into the chamber is 6nL) and so for each sample coinfection could be detected. But to confirm our findings, we use to perform Nested PCR or Realtime PCR targeted another gene then the one used into the microfluidic chip and so sometimes coinfection failed to be confirmed because of the difference in term of sensitivity of nested PCR comparatively to our technic (for more detail see Michelet et al, 2014).

Round 2

Reviewer 3 Report

I read carefully the replies of the authors to my comments. Some answers are convincing and the others not, but I can understand the challenges. For example, phylogenetic analysis is crucial. But, anyhow, in my opinion the manuscript is deserved for publication. 

Author Response

I read carefully the replies of the authors to my comments. Some answers are convincing and the others not, but I can understand the challenges. For example, phylogenetic analysis is crucial. But, anyhow, in my opinion the manuscript is deserved for publication. 

Answer: We thank the reviewer for is valuable comment. But unfortunately as explain, this time we were not able to obtain several sequence for the identified pathogen and then performed phylogenetic analysis. This step will be performed for our next experiment/publication. A sentence has been Added into the discussion section, to explain phylogenetic analysis is crucial for next time. "In future studies, phylogenetic analysis targeting several genes for pathogenic and non-pathogenic microorganisms will allow us to better answer to this question."

Table 1 was also improved.

Regards